# Comparative Analysis of Water Quality Applying Statistic and Machine Learning Method: A Case Study in Coyuca Lagoon and Tecpan River, Mexico

Humberto Avila-Perez [1,†], Enrique J. Flores-Munguía [2,†], José L. Rosas-Acevedo [2,†], Iván Gallardo-Bernal [3,†] and Tania A. Ramirez-delReal [4,*]

1  Higher School in Sustainable Development, Universidad Autonoma de Guerrero, Tecpan 40900, Mexico
2  Regional Development Science Center, Universidad Autonoma de Guerrero, Acapulco 39640, Mexico
3  Government and Public Management Faculty, Universidad Autonoma de Guerrero, Chilpancingo 39470, Mexico
4  CONACyT-CentroGeo, Centro de Investigación en Ciencias de Información Geoespacial AC, Aguascalientes 20312, Mexico
*  Correspondence: tramirez@centrogeo.edu.mx; Tel.: +52-449-1291237
†  These authors contributed equally to this work.

**Abstract:** The water quality monitoring of lotic and lentic ecosystems allows for informing the possible use in human activities and the consumption of the vital liquid. This work measures the biochemical parameters in Coyuca Lagoon and Tecpan River, localized in Guerrero, Mexico. A comparative statistical analysis of six physicochemical factors in lentic and lotic ecosystems was carried out, finding individual pH values slightly higher for the lagoon ecosystem and lower for the river. For electrical conductivity, we find river sites with parameters lower than 500 μS/cm ideal for human use and consumption. On the contrary, in sites of the lagoon system, the conductivity was higher. As for the total hardness of the river, the values are within the Mexican standard; however, for the lagoon ecosystem, the water has a higher amount of calcium and magnesium salts and is not recommended for human consumption. For chlorides, the lagoon system exceeds the limits of regulations for human consumption; otherwise, it happens with the lotic system. The values of total alkalinity and total dissolved solids are higher for the lentic system than for the lotic one. Finally, the machine learning method shows the importance of measuring other parameters to determine the water quality, especially the salinity and calcium hardness.

**Keywords:** artificial intelligence; aquatic ecosystems; applied mathematics; punctual water condition





## 1. Background

Water quality (WQ) defines the biochemical and physical factors applying standards [1]. Therefore, a water quality assessment considers the biochemical and physical factors; the importance of good quality is so it is possible to use in human activities (consumption, agriculture, and breeding, among others [2,3]). The quality represents a water status considering parameters and the water body's pollution grade.

Water bodies are subjected to various anthropogenic pressures that alter their quality, causing deep levels of contamination [4–6]. These water systems are home to a great diversity of fauna and flora influenced by altitudinal, geological, and climatic factors and physicochemical transformations of the water [7,8]. As the most notable element of water resources, rivers have a crucial function in improving the life longing of living animals or plants [9].

These variations in water quality are the result of the combination of natural processes (weathering and soil erosion) and anthropogenic contributions, such as the discharges [10–12] of waste that come from various human activities [13].

The water's physical characteristics are those that directly impact the quality conditions and acceptability of the water [14], such as the turbidity, soluble and insoluble solids, color, smell, taste, temperature, and pH. Water can contain any element on the periodic table as a universal solvent.

As part of objective 6 of the sustainable development goals (SDGs), this work reaffirms that everyone has the right to access and dispose of and to the sanitation of water for personal and domestic consumption in an acceptable, healthy, and affordable manner.

Therefore, assessing water quality in lotic and lentic ecosystems becomes relevant for sustainable water management. The collaboration of artificial intelligence in the management of specific data is essential for projections in the medium and long term for the establishment of conservation and management strategies [15].

In this sense, the consequences of the change in ecosystems, their state of conservation, and the use of the services they generate for society, and their impact on human well-being, can compromise the well-being of future generations [16]. In this way, the relationships between the functioning of ecosystems and human well-being, the human–water link, humanistic perspectives, and the interpretation of cultural prosperity are related to human aggregation toward this type of ecosystem [17].

Good quality water is indispensable for sustainable socioeconomic development [18]. Consequently, monitoring programs that provide spatiotemporal representations and reliable estimates of water quality are necessary [19].

Various works were proposed to study water quality, and some of them utilize statistics or machine learning methods to model or predict. The work of [9] conducted a study in the Karoun River, southwest of Iran; they obtained a dataset with electrical conductivity (EC), sodium ($Na^+$), calcium ($Ca^{2+}$), magnesium ($Mg^{2+}$), orthophosphate ($PO_4^{3-}$), nitrite ($NO_{2-}$), nitrate-nitrogen ($NO_{3-}$), turbidity, and pH.

They applied diverse artificial intelligence techniques to estimate the water quality indices, particularly the least square support vector machine (LS-SVM) and multivariate adaptive regression spline (MARS). The parameters to predict were the five-day biochemical oxygen demand (BOD5) and chemical oxygen demand (COD).

In [20], the water quality of a region of Mexico, particularly Zamora, Michoacán, was studied. Their work was mainly based on a biochemical analysis and used the weighted arithmetic average method. The advancement of technology and its applications in machine learning have allowed its users to become more and more widespread, in this case, in water quality analysis.

Moreover, the use of machine learning methods helps identify other kinds of parameters to determine the water quality. In this work, we use the Spearman correlation to identify the principal features to improve machine learning methods to classify the water quality for human use.

Then, the results of a logistic regression (LR) and support vector machine (SVM) model achieved 98% accuracy, adding parameters of those marked in the standard used in Mexico. This finding allows a measure of biochemical factors to analyze the water quality for human utilization.

Despite the advantages of specific analyses (physiochemical) for evaluating the water quality in aquatic ecosystems, there are challenges related to the projection of data for the visualization of future scenarios. In this context, we are testing the applicability of machine learning models as a tool to incorporate into this type of study. In addition, this type of support could reduce the costs by implementing low-end devices for data collection and processing.

The study area of the Tecpan River is located at the extreme coordinates of 17° and from 08' to 17' north latitude and 100° from 36' to 39' west longitude. Moreover, the Coyuca Lagoon is at 16° and from 56' to 57' north latitude and from 099°58' to 100°08' west longitude. The Tecpan River and the Coyuca Lagoon are bodies of water impacted by human activities on different scales. We consider the comparison of physicochemical

factors important as a mechanism that provides information for both ecosystems' protection, conservation, and management, describing the background of this work and its aims.

This paper is organized as follows: The Methods introduce the materials and methods. Then, the performance of the computational experiments for the machine learning methods is shown, and the results are presented. Finally, some concluding remarks are given. Furthermore, some ideas on the importance of coming works are outlined.

## 2. Methods

### 2.1. Data

For the Coyuca Lagoon, four sampling stations were established in the lagoon perimeter (Paraíso de los Manglares, El pedregoso, Base Aérea, and La Barra). The samples were taken fortnightly on the shore for six months, from May to October, and all stations were georeferenced (see Figure 1 and Table 1).

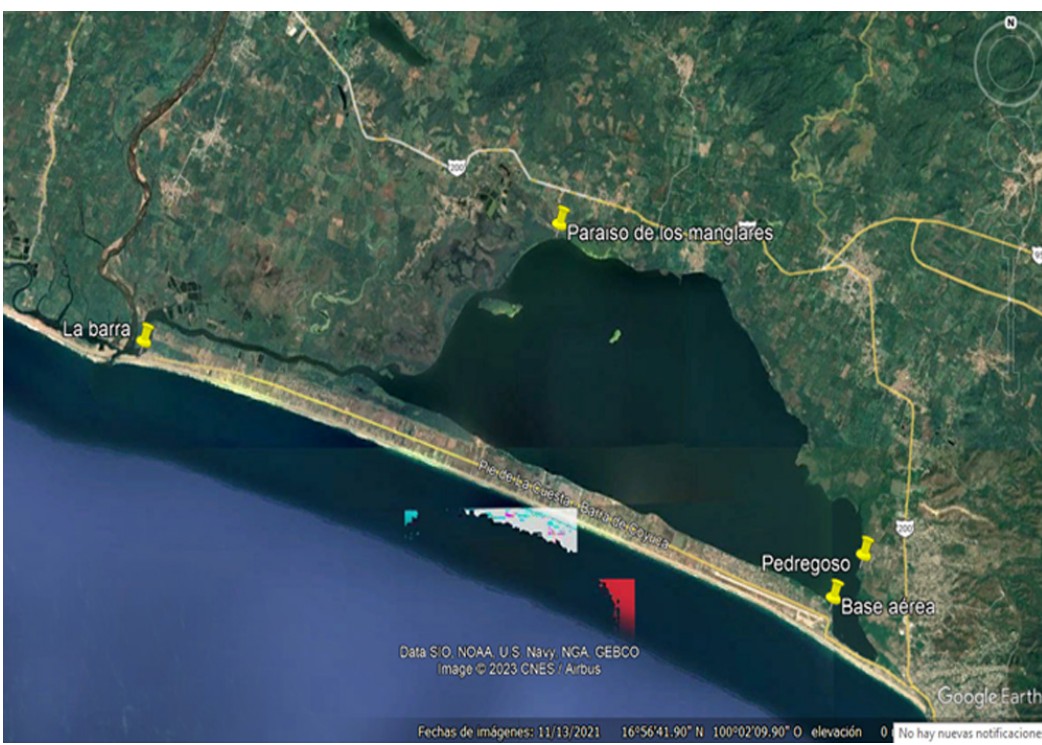

**Figure 1.** Geographical coordinates of sampling points in the Coyuca Lagoon.

**Table 1.** Location and coordinates of sampling stations in the Coyuca Lagoon and the Tecpan River.

| Station | Latitude | Longitude | Altitude |
|---|---|---|---|
| Paraiso de los Manglares | 16°57′59″ N | 100°01′44″ W | 7 masl |
| Pedregoso | 16°55′05″ N | 099°58′23″ W | 6 masl |
| Base Aérea | 16°54′41″ N | 099°58′58″ W | 7 masl |
| La Barra | 16°56′58″ N | 100°06′53″ W | 1 masl |
| Boca Chica | 17°08′18″ N | 100°38″ 46″ W | 9 masl |
| Tetitlan | 17°09′03″ N | 100°39′08″ W | 11 masl |
| Puente libramiento | 17°12′02″ N | 100°38′15″ W | 15 masl |
| Puente roto | 17°13′24″ N | 100°38′07″ W | 20 masl |
| Puente prepa | 17°14′06″ N | 100°37′32″ W | 29 masl |
| Pozumiche | 17°15′35″ N | 100°37′45″ W | 43 masl |
| El verde | 17°17′14″ N | 100°36′40″ W | 94 masl |
| El paraje | 17°18′02″ N | 100°36′19″ W | 108 masl |

For the Tecpan River, eight sampling stations were established (Boca Chica, Tetitlan, Puente Libramiento, Puente Roto, Puente Prepa, Pozulmiche, El verde, and El Paraje), and the samples were taken on the banks of the river for the same period, and all stations were georeferenced (see Figure 2 and Table 1).

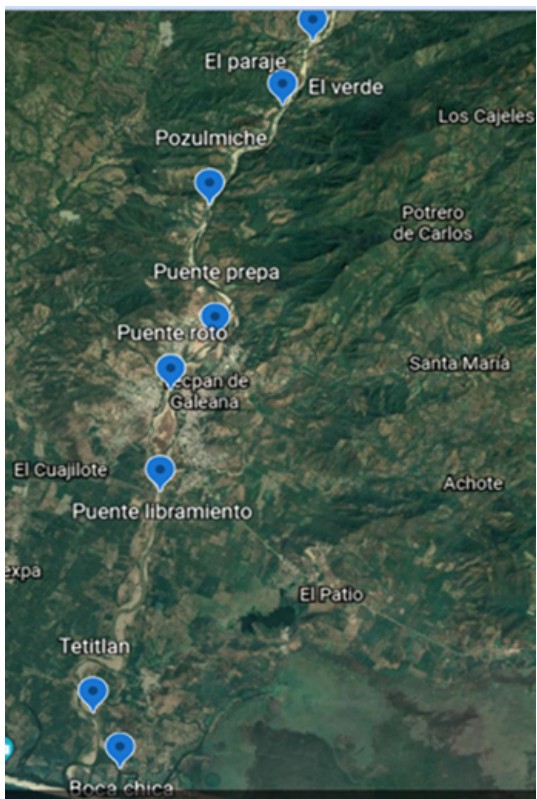

**Figure 2.** Geographical coordinates of sampling points in the Tecpan River.

The representative area was selected in each sampling station following the recommendations [21]. The bottles were duly labeled and preserved in coolers for transport to the CCDR Water Laboratory, where six physicochemical parameters were analyzed according to the Official Mexican Standards (see Table 2).

**Table 2.** Physicochemical parameters measured.

| Physicochemical Parameter | Analytical Method | Official Mexican Standard |
|---|---|---|
| pH | Potentiometric | NMX-AA-008-SCFI-2000 |
| Electrical conductivity (µS/cm) | Potentiometric | NMX-AA-093-SCFI-2000 |
| Total alkalinity | Volumetric (acid base) | NMX-AA-036-SCFI-2001 |
| Total hardness | Volumetric (complexometry) | NMX-AA-072-SCFI-2001 |
| Chlorides | Volumetric (argentometric) | NMX-AA-073-SCFI-2001 |
| Total dissolved solids | Gravimetry | NMX-AA-034-SCFI-2001 |

For the analysis and determination of water quality, a statistical package was used to compare the interrelationships presented, and the maximum permissible limits of NOM-127-SSA1-1994 (see Table 3), which is a standard used to evaluate quality, were compared to water for human consumption, as well as the Ecological Criteria for Water Quality CE-CCA-001/89.

**Table 3.** Maximum permissible limits of NOM-127-SSA1-1994.

| Characteristic | Maximum Limit |
| --- | --- |
| Chlorides | 250 mg/L |
| Total hardness (TH) | 500 mg/L |
| pH | 6.5–8.5 |
| Total dissolved solids (TDS) | 1000 mg/L |
| Electrolytic conductivity | - |
| Total alkalinity | - |

*2.2. Physicochemical Parameters*

The interpretation of the physicochemical variables estimates the quality indices and water contamination indices, when representing the valued parameters, allowing evaluation of the water quality. The National Sanitation Foundation (NSF) used nine parameters; the European Community developed the universal water quality index based on twelve variables [22]. In Peru, two indicators were applied in which, in addition to the NSF parameters, electrical conductivity, chlorides, and ammoniacal nitrogen are considered [23].

According to [24], there are at least 30 commonly used water quality indices globally, considering 3 to 72 variables. Practically all of these indices include at least 3 of the following parameters: oxygen demand (OD), biochemical oxygen demand (BOD), or chemical oxygen demand (COD), nitrogen in the form of ammonia and nitrates ($NH_{3-}N$ and $NO_{3-}N$), phosphorus in the form of orthophosphate ($PO_{4-}P$), pH, and total solids (TS).

This work presents a statistical and comparative analysis of six physicochemical variables: pH; the total hardness that considers implicit calcium hardness and magnesium hardness; electrical conductivity; chlorides; total alkalinity that includes bicarbonates; and total dissolved solids, which according to the literature, the indices must include at least three parameters.

2.2.1. pH

A measurement used to assess the acidity or alkalinity of a solution [25]. pH water for human consumption [26] should be between 6.5 and 8.5 (neutral and slightly alkaline) and from 6.0 to 9.0 according to NOM-001-SEMARNAT-2021. Waters with a pH of less than 6.5 are corrosive due to carbon dioxide, acids, or acid salts in the solution.

In general, the pH of the water does not present significant variations and is around neutrality. The problem is in wastewater or industrial discharges that can give extreme pH values. The pH varies as a function of temperature; if it increases, the pH decreases and tends to acidity. It can also vary depending on salinity, pressure or depth, and aquatic organisms' vital activity.

2.2.2. Total Hardness

The total hardness of the water or the sum of the individual hardnesses due to calcium and magnesium ions in the form of bicarbonate (NMX-AA-072-SCFI-2001), the degree of hardness is directly proportional to the concentration of metallic salts present in water [27,28]. It depends on the soil from which they come, being able to be soft or hard water. Water with less than 75 mg/L of $CaCO_3$ is considered soft; between 75 and 150 mg/L, it is moderately hard; from 150 to 300 mg/L, it is hard; and more than 300 mg/L, it is tough.

The authors of [29] adopted 100 mg/L of $CaCO_3$ as the maximum desirable concentration and 500 mg/L as the maximum admissible concentration. People generally tolerate up to 500 mg/L, the guideline value established by the World Health Organization (WHO).

2.2.3. Electrical Conductivity

This parameter refers to the ions' ability to conduct electrical current in a solution. Pure water behaves as an electrical insulator, with the substances dissolved in it providing the water with the ability to conduct electric current. It is determined by electrometry with

a conductometric electrode, expressing the result in microsiemens. Pure water is a poor conductor of electricity because its ionization ability is limited. The more ions are present in the water, the higher its conductivity.

The conductivity of water depends on the concentration and nature of the ions dissolved in it and the temperature. Usually, an increase in salts implies an increase in conductivity. The value established as usual is between 100 and 1000 μS/cm [30], although at some point it can be exceeded naturally [31].

### 2.2.4. Chlorides

They are salts resulting from chlorine gas with a metal, such as sodium. The chloride ion is incorporated into surface waters through atmospheric deposition by weathering sedimentary rocks, and discharges from industrial and wastewater in urban areas [31] are mainly associated with sodium ions [32]. Calculus kidney formation is related to the salinity and hardness of the water due to the combination of salts and calcium [29].

### 2.2.5. Total Alkalinity

According to NMX-036-SCFI-2001, the amount of solid acid necessary to reduce or neutralize the pH to 4.3 [33,34]. This parameter is not of significant health importance, but it generates rejection due to the bad taste in high concentrations.

### 2.2.6. Total Dissolved Solids

The set of total solids is defined as the remaining material that is obtained in the form of residue after subjecting the water to an evaporation process between 103 and 105 °C. Sedimentary, suspended, and dissolved solids are distinguished, and the total solids are the sum of all. In addition to being able to suppose the presence of foreign bodies or substances that could, in some cases, not be recommended, these solids increase the turbidity of the water and decrease its quality [35].

### 2.2.7. Salinity

A measure of the number of dissolved salts in water. Salts accumulate due to flooding in low-lying areas, high evaporation, plant transpiration, and, in many cases, the proximity of groundwater that can reach the surface and become salinized due to low rainfall and poor management of the water—irrigation water and fertilizers [36]. The saline content of many lakes, rivers, or streams is so tiny that these waters are called freshwater. The salt content in drinking water is, by definition, less than 0.05%. If not, the water is marked as brackish or saline [37].

### 2.2.8. Calcium Hardness

Calcium is an element present in water closely linked to its hardness. It is naturally present in surface waters due to the weathering of rocks and minerals, especially gypsum and limestone, and industrial discharges may also contribute to the increase in the concentration of this cation [31].

### 2.2.9. Magnesium Hardness

Magnesium, like calcium, is a parameter related to the hardness of the water. Magnesium is also found naturally due to the weathering of the rocks [31].

### 2.2.10. Bicarbonates

The presence of bicarbonates influences the hardness and alkalinity of the water. Its presence in freshwater can occur naturally by the dissolution and weathering of rocks or by contributions from industrial discharges. Bicarbonates are present in waters with a pH value between 6.5 and 8.5 [31].

### 2.3. Analysis Techniques

#### 2.3.1. Analysis of Variance and Tukey Test

The analysis of variance (ANOVA) is the basis of the experimental design or statistical methodology oriented to the planning and analyzing of an experiment or tests that verify the validity of the hypotheses established about the causes of a particular problem or that affects a specific object variable of study [38,39].

As an aid to these designs, studying variance in its most basic form is to check whether a variable with several levels, called independent, can explain the variations observed in one or more, which others call dependent [38,40].

The operation of the ANOVA technique is, broadly speaking, as follows: In order to compare the means of Y associated with the different levels of the factor $(X_1, X_2, \ldots, X_n)$, we will compare a measure of the variation between different levels (MS-factor) with a measure of the variation within each level (MS-error). If the MS-factor is significantly greater than the MS-error, we will conclude that the means associated with different factor levels are different. It means that the factor significantly influences the dependent variable Y. If, on the other hand, the MS-factor is not significantly more significant than the MS-error, we will not reject the null hypothesis that all the means associated with different levels of the factor coincide [41].

Moreover, the Tukey test is used, which aims to compare the individual means from an analysis of variance of several samples subjected to different treatments; recommended for its simplicity because it allows all possible comparisons to be made two by two and because it has confidence limits [42,43]. It also allows for discerning whether the results obtained are significantly different.

#### 2.3.2. Spearman Coefficient Analysis

The correlation structure between the variables is studied using the Spearman coefficient. Spearman correlation is a widely used metric [44], competent in estimating linear and nonlinear connections; this coefficient is calculated, and the examination establishes the constant lowering or growth in the values of a variable, defined as monotonically increasing or decreasing [45].

#### 2.3.3. Logistic Regression

The logistic regression method specifies the relation between dependent and independent variables. The employment of this technique is beneficial when the dependent variable can be observed. The model is provided for the practical answers using the maximum likelihood approach, then the best match is predicted [46].

#### 2.3.4. Support Vector Machine

Ref. [47] presented a supervised machine learning technique called support vector machines. SVM employs kernel procedures to separate classes by support vectors; the input interprets the training data to transform it into an upper-dimensional space. Then, a hyperplane is divided to obtain the correspondences of the classes.

#### 2.3.5. Performance Metric

The performance of machine learning algorithms is measured by applying some metrics; in this work, accuracy, precision, and recall [48].

Accuracy is the expression for all the samples classified correctly between all samplings, and Equation (1) represents the mathematic estimation.

$$\text{Accuracy} = \frac{TP + TN}{TP + TN + FP + FN} \tag{1}$$

Precision is the relation between the positive samples categorized correctly and the sum of all samples classified as positive, and Equation (2) defines the mathematic computation.

$$\text{Precision} = \frac{TP}{TP + FP} \tag{2}$$

The recall is the relation between the positive samples categorized correctly and the sum of valid positive samples, and Equation (3) describes this ratio.

$$\text{Recall} = \frac{TP}{TP + FN} \tag{3}$$

where $TP$ is true positive, $TN$ is true negative, $FP$ is false positive, and $FN$ is false negative.

### 2.4. Experimental Setup

The machine learning techniques used in this work are programmed in Python 3.7 and the sklearn library; these techniques are employed to corroborate the water quality classification, selecting the features by Spearman correlation.

The data used are acquired in all the samples in the eight parameters presented in the above section.

### 3. Results

The physicochemical values were compared considering the permissible limits of the NOM-127-SSA1-1994 (see Table 4) found for the Coyuca Lagoon in which the total hardness, chlorides, and total dissolved solids exceed the limits of the norm. On the contrary, in the Tecpan River, only the pH exceeds the permissible limits.

On the contrary, in the Tecpan River, only the pH is slightly below the permissible limits. It can be recommended for agricultural irrigation and human consumption after purification treatment because the other parameters, such as the total hardness, chlorides, and solids, are in the range according to the standards.

**Table 4.** Maximum permissible parameters.

| Ecosystem | Place | pH | Conductivity | TH | Chlorides | Alkalinity | TDS |
|-----------|-------|----|--------------|----|-----------|------------|-----|
| Lagoon | Paraiso manglares | 6.6 | 6265 | 1008 | 1832 | 670 | 3099 |
| | Pedregoso | 7.0 | 7060 | 1366 | 1958 | 761 | 3291 |
| | Base aérea | 6.8 | 6429 | 953 | 1719 | 465 | 3091 |
| | La barra | 6.6 | 1752 | 479 | 344 | 395 | 1494 |
| River | Boca chica | 6.2 | 318 | 249 | 77 | 236 | 161 |
| | Tetitlan | 6.2 | 141 | 124 | 44 | 99 | 75 |
| | Puente libramiento | 5.9 | 241 | 232 | 67 | 213 | 124 |
| | Puente roto | 6.1 | 107 | 97 | 29 | 96 | 52 |
| | Puente prepa | 6.0 | 102 | 86 | 27 | 83 | 51 |
| | Pozulmiche | 6.1 | 128 | 96 | 31 | 93 | 64 |
| | El verde | 6.2 | 110 | 103 | 32 | 95 | 59 |
| | El paraje | 5.9 | 119 | 103 | 35 | 95 | 59 |

*Physicochemical Parameters*

The pH values registered in the Tecpan River and in the Coyuca Lagoon are shown in Table 4. The lagoon with values between 6.60 to 7.01 is within the permissible limit of 6.5–8.5 according to NOM-127-SSA1-1994 [49], with a slight tendency to alkalinity in the lagoon system, similar to that reported by [50] in Laguna de Chautengo, Guerrero. On the contrary, in the Tecpan River, values from 5.95 to 6.26 were found, showing a slight tendency to an acidic pH in the Libramiento and Paraje sites, similar to those reported by [51], mentioning that in the Tres Palos Lagoon, the pH can increase if there is much water discharged of untreated wastewater and organic matter or it may decrease due to low concentrations of dissolved oxygen.

Moreover, the pH variability can be impacted by the water treatment applied or the type of basin from which it comes from, the minerals' richness, given that the soil is acidified [52].

It results in water reactions with $Al^{3+}$, $Fe^{2+}$, $Mn^{2+}$, and $NO_{3-}$, which release $H^+$ into the solution [53,54], altering the potential of hydrogen in the water.

We can see slightly higher individual values for the lagoon ecosystem at the Air Base, La Barra, and El Pedregoso stations; on the contrary, for the lotic ecosystem of the Tecpan River, the lowest values were reported at the El Paraje and Puente Libramiento sites, as we can see in Figure 3.

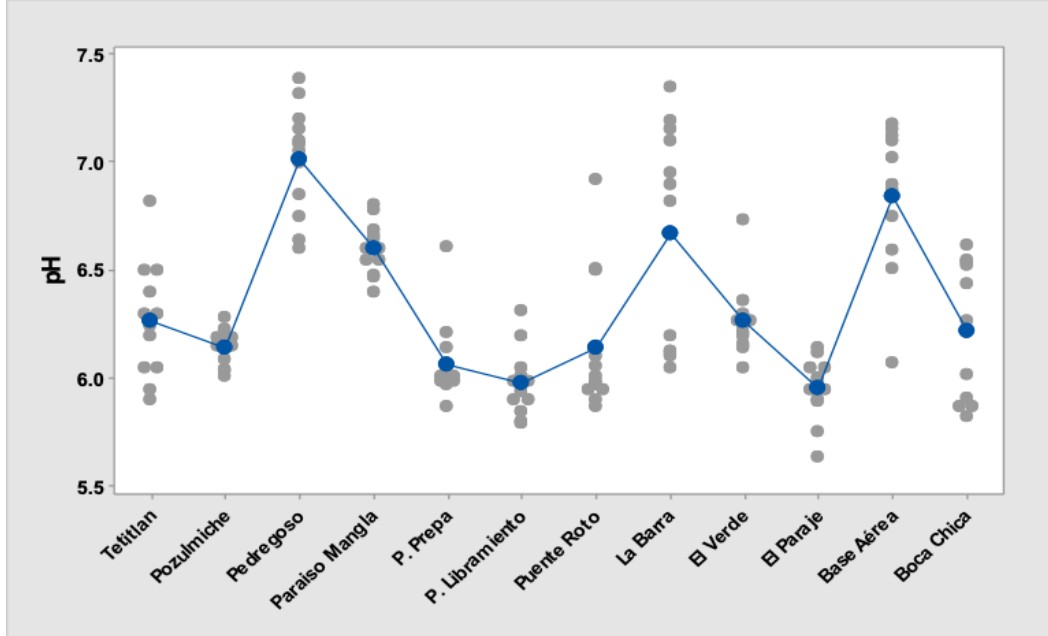

**Figure 3.** pH values in each location; the grey points refer to the data dispersion in relation to the mean represented by the blue line.

An analysis of the information was carried out (ANOVA in Mini Tab 17) (Table 5) using the option boxes and whiskers by the Tukey method that considers the grouped means, obtaining a comparison in pairs where the conformation of groups with similar means in sites is appreciated.

In Table 5, the means that do not share a letter are significantly different, and the means that share a letter have certain similarities in such a way that groups with similar means were formed. Group AB: the "Base Aérea" and "La Barra" sites have similarities in the lagoon ecosystem sites, group BC: the "Paraíso de los manglares" site, group CD: the "Tetitlan" and "El Verde" sites. The sites of the letters A and D are significantly different. The sites of group D correspond to the river that must be purified for human use.

The results obtained in six months of sampling show that the electrical conductivity is a numerical expression that indicates the capacity of a solution to transport an electric current and gives us an idea of the degree of mineralization of natural, potable, residual, treated residual, and processed water (NMX-AA-093-SCFI-2000).

There are no reference values that can be used as the maximum permissible limit; therefore, according to [51], it is recommended to compare and relate to the salinity, total dissolved solids (TDS), and biochemical oxygen demand (BOD).

The comparison of this study can be seen in Table 6, where we find that the sites located in the river with ranges lower than 318 μS/cm with little saline influence and values lower than 500 μS/cm are ideal for human use and consumption. On the contrary, sites in the lagoon system, due to the proximity to the sea, the deposition of organic matter, and the

discharge of untreated wastewater, have higher conductivity due to the concentrations of sodium chloride and chloride ions, similar to that reported by [51].

**Table 5.** Tukey's pairwise comparisons, grouping of information using the Tukey method confidence of 95%.

| Factor | N | Mean | Pair |
|---|---|---|---|
| Pedregoso | 12 | 7.0108 | A |
| Base aérea | 12 | 6.8425 | A B |
| La barra | 12 | 6.672 | A B |
| Paraíso mangle | 12 | 6.6025 | B C |
| Tetitlan | 12 | 6.2675 | C D |
| El Verde | 12 | 6.2650 | C D |
| Boca chica | 12 | 6.2200 | D |
| Puente Roto | 12 | 6.1442 | D |
| Pozulmiche | 12 | 6.1400 | D |
| Puente Prepa | 12 | 6.0675 | D |
| Punte Libramiento | 12 | 5.9775 | D |
| El Paraje | 12 | 5.9525 | D |

**Table 6.** Mean electrical conductivity.

| | Place | Mean |
|---|---|---|
| River | Boca Chica | 318 μS/cm |
| | Tetitlan | 141 μS/cm |
| | Puente libramiento | 241 μS/cm |
| | Puente roto | 107 μS/cm |
| | Puente prepa | 102 μS/cm |
| | Pozulmiche | 128 μS/cm |
| | El verde | 110 μS/cm |
| | El paraje | 119 μS/cm |
| Lagoon | Paraiso de los manglares | 2459 μS/cm |
| | El Pedregoso | 2322 μS/cm |
| | Base aérea | 2253 μS/cm |
| | La barra | 1831 μS/cm |

Figure 4 shows the total hardness values for the two ecosystems found. For the Tecpan River, the values are within the norm; however, for the lagoon ecosystem, in sites such as El pedregoso, Paraiso de los manglares, La Barra, and the Base Aérea, the water is hard and is not recommended for human consumption.

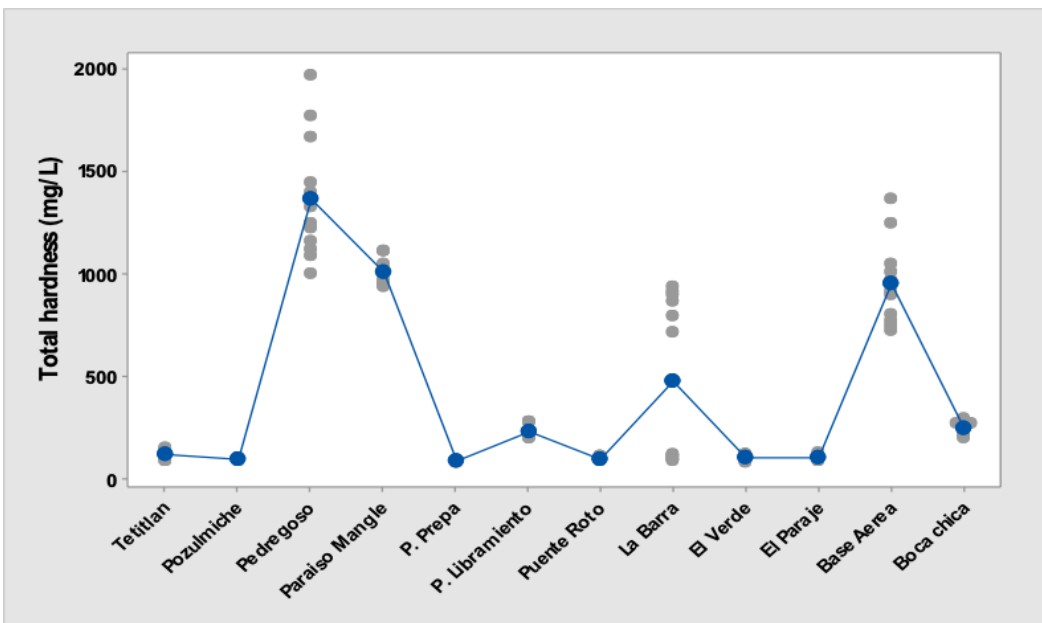

**Figure 4.** Total hardness values in each location; the grey points refer to the data dispersion in relation to the mean represented by the blue line.

In Mexico, there is the NOM-127-SSA1-1994 regulation that indicates the permissible parameters of water for human consumption, establishing a value of 250 mg/L for the concentration of chlorides. Figure 5 shows the comparison of the two environmental systems where we can see that the lagoon system exceeds the limits of the NOM; on the contrary, the lotic system of the Tecpan River is within the permissible limits for human consumption.

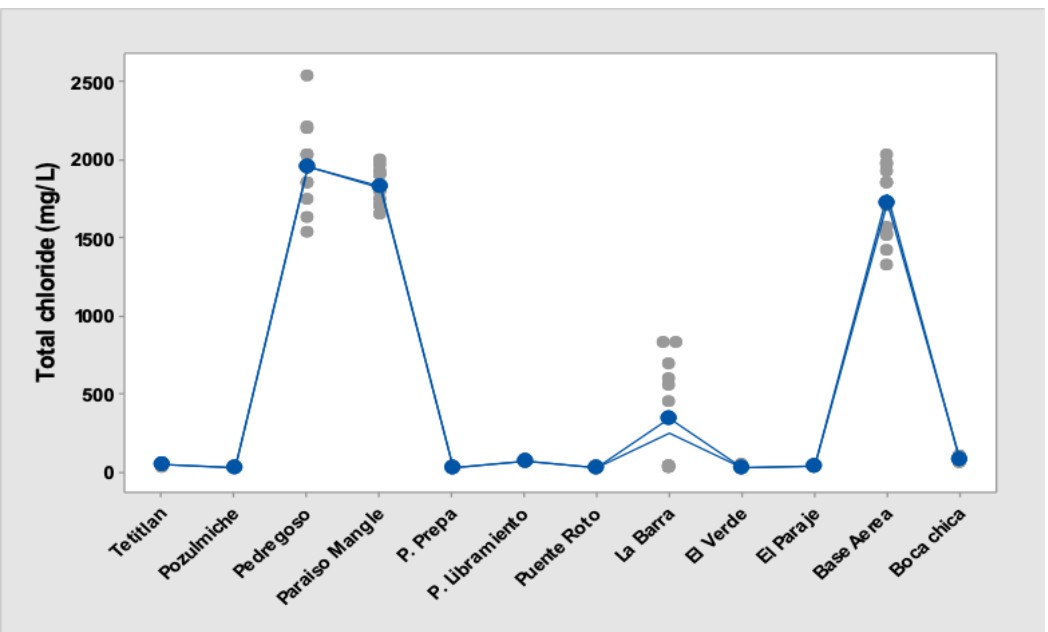

**Figure 5.** Chlorides values in each location; the grey points refer to the data dispersion in relation to the mean represented by the blue line.

It is natural to find values from 200 to 500 mg/L. Specifically for the lagoon ecosystem, we found higher total alkalinity values, unlike the Tecpan River, where we found values of up to 500 mg/L, as shown in Figure 6.

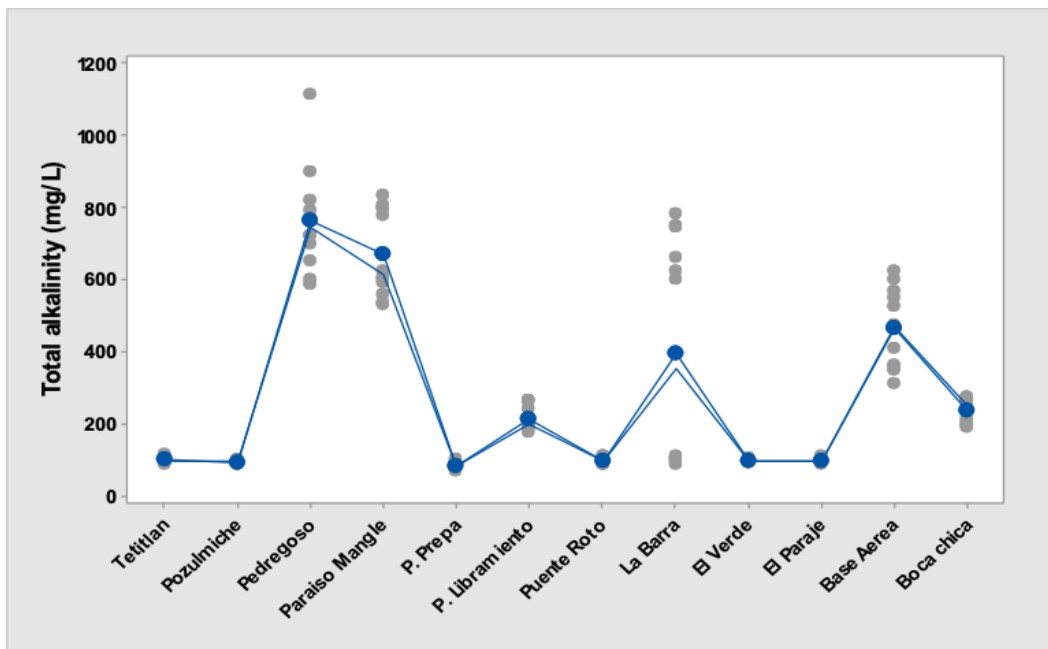

**Figure 6.** Total alkalinity values in each location; the grey points refer to the data dispersion in relation to the mean represented by the blue line.

Water is a means of transport that contains a variety of solid materials such as sand, clay, silt, and other loose soil particles from erosion [55] or from the decomposition of plants and animals that gives it high turbidity, which is dragged from the upper part of the basin. In aquatic environments, particulate matter transports chemical compounds from the water column to the bottom sediments [56]. Specifically, Figure 7 of the TDS shows us that in the lagoon ecosystem, the highest values of these solids were presented due to the composition and structure of the lentic system (2743 mg/L), unlike the lotic ecosystem of Tecpan, which presented low values of the TDS (80 mg/L).

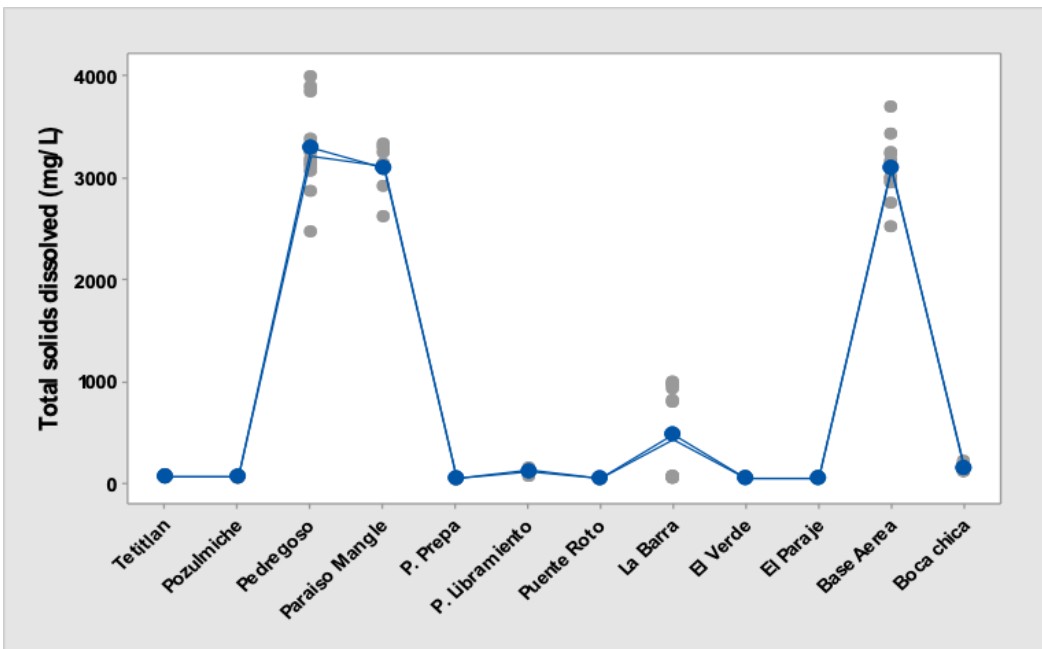

**Figure 7.** Total dissolved solids values in each location; the grey points refer to the data dispersion in relation to the mean represented by the blue line.

Applying the Spearman correlation coefficient, the fourth main features selected are the salinity, conductivity, total dissolved solids, and total hardness; the above is directly related to the water quality according to the lotic or lentic system.

These four parameters are used to train the logistic regression and support vector machine models; the samples are divided into 80% for the train and 20% for the test. The accuracy for the logistic regression is 96.55%, a precision of 96.72%, and a recall of 96.55%. The best performance is in the accuracy, precision, and recall for the SVM, obtaining 100% in the three metrics.

## 4. Discussions

The statistical projection of the data obtained allows us to quickly know the conditions of the water systems in the study areas. The alkalinity is an important indicator that provides information on the atmospheric carbon sequestration ratios and the inherent chemical ratios in the water and soil.

Anthropogenic activities are a primary factor in the disturbances in the quality of these natural sources. The increase in atmospheric $CO_2$ has caused a considerable increase in the acidification of hydrological systems. Mathematical and statistical projections are powerful and low-cost tools that make projections of the current and future conditions of the quality of hydrological bodies.

This exploratory research presents an approximation of the current conditions of the quality of the lotic and lentic systems studied in this work, mainly in the relationship between the pH and alkalinity concentrations. It is suggested to carry out more monitoring, expanding the analytical spectrum to find out other causes that could impact the decrease in pH and its relationship with capturing atmospheric carbon.

Lentic systems do not present a constant unidirectional flow due to the volume retention of water that allows the solids' precipitation and nutrient enrichment; the above determine their trophic status. This explains the values found in the Laguna de Coyuca under the Mexican regulations (NOM-127-SSA1-1994) where the permissible limits are exceeded in the electrical conductivity, total hardness, chlorides, alkalinity, and total dissolved solids parameters. Therefore, previous purification treatment is recommended for its use. Otherwise, in the lotic ecosystem of Tecpan, it occurred.

It should be noted that the determining factors in altering the taste of drinking water correspond to the level of the hardness and alkalinity present in the water due to the dragging of minerals. The total hardness and the calcium and magnesium presence determine the potable water, and they must be monitored because, in excess, they can drive cardiovascular problems.

The total alkalinity influences the pH due to the acidification processes in the lagoon ecosystem (underneath the permissible limit) and affects the potability for human consumption (pH 6.5). The punctual analysis of the physical, chemical, and biological properties of both the lotic and lentic ecosystems, as well as the data projection in future scenarios, are based on scientific information that, together with the application of machine learning models, provides an alternative low-cost means of processing data on these types of ecosystems that are increasingly impacted and require conservation.

Likewise, artificial intelligence methods can remarkably reduce water supply and sanitation systems costs and help ensure compliance with drinking and wastewater treatment quality. Therefore, modeling and predicting water quality to control water pollution has been widely researched.

## 5. Conclusions

The parameter with the most significant influence in both ecosystems for the water quality is the total dissolved solids, attributable to the dragging of solids from the upper parts of the watershed, especially in the rainy season.

The sites with the highest physicochemical values in both ecosystems are located very close to human settlements, in the Coyuca Lagoon and the Tecpan River, and the Boca

Chica (with higher conductivity) and Puente Roto sites. In the river, the parameters were determined within the regulations (electricity conductivity, total hardness, chlorides, total alkalinity, and total dissolved solids), but not for the lagoon system.

Regarding the pH of the river, we found slight acid values for the Puente Libramiento site. This is consistent with the existence of a sewage treatment plant that is not in operation, and municipal discharges are not treated.

The decrease in pH is associated with the use of phosphorous fertilizers, which can lower the pH of the soil, and with the removal of sediment in the rainy season, the pH of the water also tends to rise downstream.

The total hardness, total alkalinity, and total dissolved solids for the Tecpan River were three slightly high parameters for the Puente Libramiento and Boca Chica sites, probably caused by tourist activity in both places and the existence of a treatment plant that is not in operation.

This work demonstrates that machine learning techniques are helpful in training models to predict or classify water quality; the support vector machine (SVM) has the best performance.

**Author Contributions:** Conceptualization, J.L.R.-A. and H.A.-P.; methodology, H.A.-P.; software, E.J.F.-M. and T.A.R.-d.; validation, T.A.R.-d.; formal analysis, E.J.F.-M. and T.A.R.-d.; investigation, J.L.R.-A., H.A.-P., T.A.R.-d. and I.G.-B.; resources, J.L.R.-A. and H.A.-P.; data curation, E.J.F.-M., H.A.-P. and T.A.R.-d.; writing—original draft preparation, J.L.R.-A., H.A.-P., T.A.R.-d and I.G.-B.; writing—review and editing, J.L.R.-A., H.A.-P., T.A.R.-d. and I.G.-B.; visualization, H.A.-P. and T.A.R.-d.; supervision, J.L.R.-A. and H.A.-P.; project administration, J.L.R.-A. and H.A.-P. All authors have read and agreed to the published version of the manuscript.

**Funding:** The authors received no financial support for the research, authorship, and/or publication of this article.

**Data Availability Statement:** Publicly available datasets were analyzed in this study. This data can be found here: https://github.com/taniaglae/water_quality (generated on 30 May 2022).

**Acknowledgments:** The authors thank Universidad Autónoma de Guerrero, through the Center for Regional Development Sciences (especially the Environmental Chemistry Laboratory), in Acapulco, for the facilities provided.

**Conflicts of Interest:** The authors declare no potential conflict of interest with respect to the research, authorship, and/or publication of this article.

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
