# Peer review of "Comparative Analysis of Water Quality Applying Statistic and Machine Learning Method: A Case Study in Coyuca Lagoon and Tecpan River, Mexico"

_water, doi:10.3390/w15040640_

Round 1

Reviewer 1 Report

1.     The authors should read the guide for authors.

2.     Please correct affiliation 2.

3.     Some keywords can be found in the title, and should be replaced with other words that increase the visibility of the article.

4.     What were the standards which had been followed through the water sampling?

5.     Please explain what IoT means where it first appears.

6.     The writing needs revisions in terms of editing. You should pay more attention to indices of chemical formulas and oxidation numbers (eg. L41, L115, L116, L137, L139, L270).

7.     In Table 5, please move the values in the last two rows to the right.

8.     Only six physicochemical parameters were analyzed, but they aren’t enough to evaluate the water quality. For the future, the authors should consider to determine the biochemical parameters COD, BOD5, nitrates, heavy metals and microbiological parameters to evaluate the water quality.

9.     In section “Author Contributions”, please write which is the contribution of each author.

10.  The English needs to be improved, maybe you can turn to a professional proofreading service.

Author Response

We would like to thank the anonymous reviewers

for their helpful comments as well as the editor for their time. These comments helped us to improve our manuscript.

  1.     The authors should read the guide for authors.

Thank you for the observation; we have corrected the template.

  1.     Please correct affiliation 2.

Thank you for your observation; the affiliation has been corrected.

  1.     Some keywords can be found in the title, and should be replaced with other words that increase the visibility of the article.

Thanks in advance; the keywords have changed to Artificial intelligence; aquatic ecosystems; applied mathematics; punctual water condition.

  1.     What were the standards which had been followed through the water sampling?

Thanks for the observation; we followed the Mexican Standards in Table 2.

  1.     Please explain what IoT means where it first appears.

Thank you for the recommendation; we corrected it.

  1.     The writing needs revisions in terms of editing. You should pay more attention to indices of chemical formulas and oxidation numbers (eg. L41, L115, L116, L137, L139, L270).

Thank you for the recommendation; we corrected themt.

  1.     In Table 5, please move the values in the last two rows to the right

Thanks for the observation; the Table has been corrected.

  1.     Only six physicochemical parameters were analyzed, but they aren’t enough to evaluate the water quality. For the future, the authors should consider to determine the biochemical parameters COD, BOD5, nitrates, heavy metals and microbiological parameters to evaluate the water quality.

Thank you in advance; we extended the discussion section with the idea that this research was exploratory; the physicochemical parameters performed were correlated with a qualitative analytical method that preliminarily evaluated the quality of the water bodies. According to the results obtained, it is considered to carry out more monitoring in the study areas that contemplate more physicochemical analytical methods (biochemical oxygen demand, chemical oxygen demand, heavy metals) and microbiological (pathogens) to obtain more excellent knowledge about hydrological quality in which they are.

  1.     In section “Author Contributions”, please write which is the contribution of each author.

Thank you for the observation, the author contribution section is completed.

  1. The English needs to be improved, maybe you can turn to a professional proofreading service.

Thank you for your advice; we carefully review the grammar and use Grammarly software.

Reviewer 2 Report

Water quality monitoring of lotic and lentic ecosystems allows for informing the possible use in human activities and consumption of the vital liquid. This work measure biochemical parameters in Coyuca Lagoon and Tecpan River localized in Guerrero, Mexico. A comparative statistical analysis of six physicochemical factors in lentic and lotic ecosystems was carried out. Humberto Avila-Perez et al. demonstrated comparative analysis of water quality applying Statistic and Machine Learning method: a case study, which could provide information regarding to the evaluation of water pollution in the Coyuca Lagoon and Tecpan River, Mexico. Overall, the paper is well-written but still need carful revision.

Main concerns:

(1)  The abstract should be polished and improved, especially the research significance should be highlighted.

(2)  In the introduction section, the authors should include a testable hypothesis and an appropriate literature review introducing nutrients in worldwide Water quality monitoring of lotic and lentic ecosystems, especially using Machine Learning method

(3) Method section 3.2. Physicochemical parameters, the authors should integrate the parameters.

(4) I recommend separate the Results and Discussions sections, make deeply discussion and clearly description results.

(5) Conclusion should be summarized well, there are many paragraphs.

(6) There are many tables (8 Tables) in the MS. Please integrate the tables, and make clear show

(7) Author Contributions section were not shown, and added the future research significance and impacts.

(8) The manuscript could use a careful review for language and grammatical correctness.

Author Response

We would like to thank the anonymous reviewers

for their helpful comments as well as the editor for their time. These comments helped us to improve our manuscript.

Water quality monitoring of lotic and lentic ecosystems allows for informing the possible use in human activities and consumption of the vital liquid. This work measure biochemical parameters in Coyuca Lagoon and Tecpan River localized in Guerrero, Mexico. A comparative statistical analysis of six physicochemical factors in lentic and lotic ecosystems was carried out. Humberto Avila-Perez et al. demonstrated comparative analysis of water quality applying Statistic and Machine Learning method: a case study, which could provide information regarding to the evaluation of water pollution in the Coyuca Lagoon and Tecpan River, Mexico. Overall, the paper is well-written but still need carful revision.

Main concerns:

(1)  The abstract should be polished and improved, especially the research significance should be highlighted.

Thank you for the recommendation; we corrected it.

(2)  In the introduction section, the authors should include a testable hypothesis and an appropriate literature review introducing nutrients in worldwide Water quality monitoring of lotic and lentic ecosystems, especially using Machine Learning method 

Thank you for the observation; we extended the introduction to include it.

(3) Method section 3.2. Physicochemical parameters, the authors should integrate the parameters

Thank you in advance;  we integrated them.

(4) I recommend separate the Results and Discussions sections, make deeply discussion and clearly description results

Thank you for the recommendation; we have separated this section, we extended both.

(5) Conclusion should be summarized well, there are many paragraphs.

Thank you for the observation, we have summarized it. 

(6) There are many tables (8 Tables) in the MS. Please integrate the tables, and make clear show 

Thank you for the observation, now we have only six tables.

(7) Author Contributions section were not shown, and added the future research significance and impacts.

Thank you for the observation; the author contribution section is completed. Also, we added future research significance and impacts.

(8) The manuscript could use a careful review for language and grammatical correctness

Thank you for your advice; we carefully review the grammar.

Round 2

Reviewer 2 Report

After revision, the MS was improveed. However, there are 2 concerns as follows:

(1)There are many paragrahs in the introduction, the authors should review the past literatures based on their research concent and put out a testable hypothesis, especially the Novelty of the research 

(2)the in the Discussions sections, the discussion is very short, and the it didn't make deeply discussion based on results and previous studies in the introduction.

   I strongly recommend the the authors should upload the tracked revised MS, mark the revisions.  

Author Response

After revision, the MS was improveed. However, there are 2 concerns as follows:
(1)There are many paragrahs in the introduction, the authors should review the past literatures based on their research concent and put out a testable hypothesis, especially the Novelty of the research 

Thank you for your observation; we have deleted some paragraphs and added the hypothesis.
In the MS the text added is red, and the deleted are crossed. 

(2)the in the Discussions sections, the discussion is very short, and the it didn't make deeply discussion based on results and previous studies in the introduction.
  I strongly recommend the the authors should upload the tracked revised MS, mark the revisions.  

Thank you for your observation; we have added some paragraphs to the discussions.
In the MS, the text added is red in the discussions section.

Round 3

Reviewer 2 Report

Thank you for addressing all my comments so thoroughly. The manuscript has significantly improved and is now fit to be published in this journal.